# SSHR: More Secure Generative Steganography with High-Quality Revealed Secret Images

Jiannian Wang[1]    Yao Lu[1]    Guangming Lu[1]

## Abstract

Image steganography ensures secure information transmission and storage by concealing secret messages within images. Recently, the diffusion model has been incorporated into the generative image steganography task, with text prompts being employed to guide the entire process. However, existing methods are plagued by three problems: (1) the restricted control exerted by text prompts causes generated stego images resemble the secret images and seem unnatural, raising the severe detection risk; (2) inconsistent intermediate states between Denoising Diffusion Implicit Models and its inversion, coupled with limited control of text prompts degrade the revealed secret images; (3) the descriptive text of images(*i.e.* text prompts) are also deployed as the keys, but this incurs significant security risks for both the keys and the secret images. To tackle these drawbacks, we systematically propose the SSHR, which joints **the Reference Images** with the adaptive keys to govern the entire process, enhancing the naturalness and imperceptibility of stego images. Additionally, we methodically construct an **Exact Reveal Process** to improve the quality of the revealed secret images. Furthermore, adaptive **Reference-Secret Image Related Symmetric Keys** are generated to enhance the security of both the keys and the concealed secret images. Various experiments indicate that our model outperforms existing methods in terms of recovery quality and secret image security.

---
[1]Department of Computer Science and Technology, University of Harbin Institute of Technology (Shenzhen), Shenzhen, Guangdong, China. Correspondence to: Yao Lu <luyao2021@hit.edu.cn>, Guangming Lu <luguangm@hit.edu.cn>.

*Proceedings of the $42^{nd}$ International Conference on Machine Learning*, Vancouver, Canada. PMLR 267, 2025. Copyright 2025 by the author(s).

## 1. Introduction

Due to the demand of privacy, security, and data protection in the era of digital communication and AI development, steganography has emerged as an essential technology. Steganography embeds diverse types of secret messages into a container medium in an undetectable manner, which has broad applications across fields (Vyas et al., 2023; Wouters, 2024; Feng et al., 2024; Liu & Bu, 2024).

Currently, image steganography techniques are primarily divided into cover-based and generative steganography methods. Cover-based methods, such as LSBM (Mielikainen, 2006), HUGO (Pevnỳ et al., 2010), UNIWARD (Sameer & Naskar, 2018), and deep learning based methods (Baluja, 2017; 2019; Lu et al., 2021; Jing et al., 2021; Guan et al., 2022), embed secret messages within the cover images. Compared to cover-based methods, generative steganography methods generating the suitable stego images directly from the secret images via neural network without using the cover images, such as CycleGAN (Zhu et al., 2017) and encoder-decoder models (Zhou et al., 2015). Recently, the diffusion model(Ho et al., 2020) has been incorporated into the generative image steganography task, such as CRoSS (Yu et al., 2024) and DiffStega (Yang et al., 2024), with text prompts, which are the descriptive text of images, being employed to guide the entire conceal and reveal processes.

However, existing diffusion model-based generative methods still face several challenges, as illustrated in Figure 1. **Firstly**, constrained by the restricted control over specific semantic regions via text prompts, the generated stego images closely resemble the secret images, particularly in the background. Occasionally, the generated stego images seem weird and unnatural, raising the severe detection risk from third parties. **Additionally**, owing to the inconsistency of intermediate states between Denoising Diffusion Implicit Models (DDIM) (Song et al., 2021) and its inversion, coupled with the restricted control exerted by text prompts (Hertz et al., 2023; Zhang et al., 2024; Wang et al., 2024), the quality of the revealed secret images has been significantly declined, posing substantial challenges for practical applications. **Finally**, text prompts, also utilized as cryptographic keys to enhance steganography security, are vulnerable to being guessed due to their essence of descriptive text

*Figure 1.* The pipeline comparison of (a) CRoSS, (b) DiffStega, and (c) our method, with main differences highlighted in red dotted boxes. Our approach enables significant modifications to the secret image while enhancing the quality of revealed images. Additionally,We use a **Reference-Secret Image Related Symmetric Key** ($k_{sym}$) as the cryptographic key, rather than text prompt used in previous methods.

for images, creating serious security risks for both the keys and concealed secret images.

To address these problems, this paper systematically proposes a secure generative steganography method, SSHR, based on the diffusion model. Given the inadequate control exerted by text prompts in diffusion-based generative steganography models, this paper substitutes text prompts with reference images as guiding conditions. The abundant semantic information inherent in reference images facilitates superior control of stego images generation. Distinct from cover-based methods, our model treats these reference images as guides for stego images generation, rather than as simple containers for concealed secret images. An exact reveal process is performed to minimize damage to the revealed secret images. Furthermore, the adaptive keys, which will be joined with the reference images to guide the entire process, are generated to compensate for the absence of keys after removing text prompts. Experimental results indicate that our method achieves notable improvements in effectiveness and security over existing models. Our main contributions are:

- We systematically propose a novel generative steganography method joints **the Reference Images** with the adaptive keys to govern the entire process, enhancing the naturalness and imperceptibility of stego images.

- We methodically construct an **Exact Reveal Process** to precisely reverse the conceal process, diminishing the errors in the reveal process and enhancing the quality of the revealed secret images.

- We propose a **Reference-Secret Image Related Symmetric Key (RSRK)** generation module to generate the keys, enhancing the security of both the keys and the concealed secret images.

- We conduct extensive experiments to demonstrate that our model surpasses existing methods in terms of stego images quality and security, as well as the quality of the revealed secret images.

## 2. Related work

### 2.1. Cover-based Image Steganography

Traditional cover-based image steganography conceals secret data within a cover image, either in the spatial domain using techniques like Least Significant Bits (Mielikainen, 2006) or in the transform domain, such as Discrete Fourier Transform. Baluja (Baluja, 2017; 2019) initially proposed a deep learning method to conceal a full-size image within another. Universal Deep Hiding (UDH) (Zhang et al., 2020) introduced a universal pipeline for image steganography that contrasts with the pipeline in (Baluja, 2017; 2019). ISN (Lu et al., 2021) pioneered the use of Invertible Neural Networks (INN), setting a benchmark by utilizing INN's reversible properties for conceal and reveal, showcasing superior capabilities. Following this, models such as HiNet (Jing et al., 2021), DeepMIH (Guan et al., 2022), and other INN-based approaches have been proposed to further explore INN's potential in image steganography.

The reliance on cover images limits the applicability of these methods, and their stability and detection resistance need enhancement. In contrast, our model utilizes a reference image to guide stego image generation rather than serving as a container for the secret image.

### 2.2. Generative Steganography

Generative steganography eliminates the reliance on cover images by directly generating stego images from the secret images using neural networks, such as CycleGAN (Zhu et al., 2017) and encoder-decoder models (Zhou et al., 2015). Wei et al. (Wei et al., 2022) established a bi-directional mapping between stego images and secret data, integrating the generator and extractor within a single Flow-based network. Distinguished from these methods, Generative Steganography Diffusion (GSD) (Wei et al., 2023) leverages the advanced capabilities of diffusion models. Additionally, CRoSS (Yu et al., 2024) and DiffStega (Yang et al., 2024), both diffusion-based models with text prompts being employed to guide the entire process, demonstrate excellent controllability and resilience against attacks.

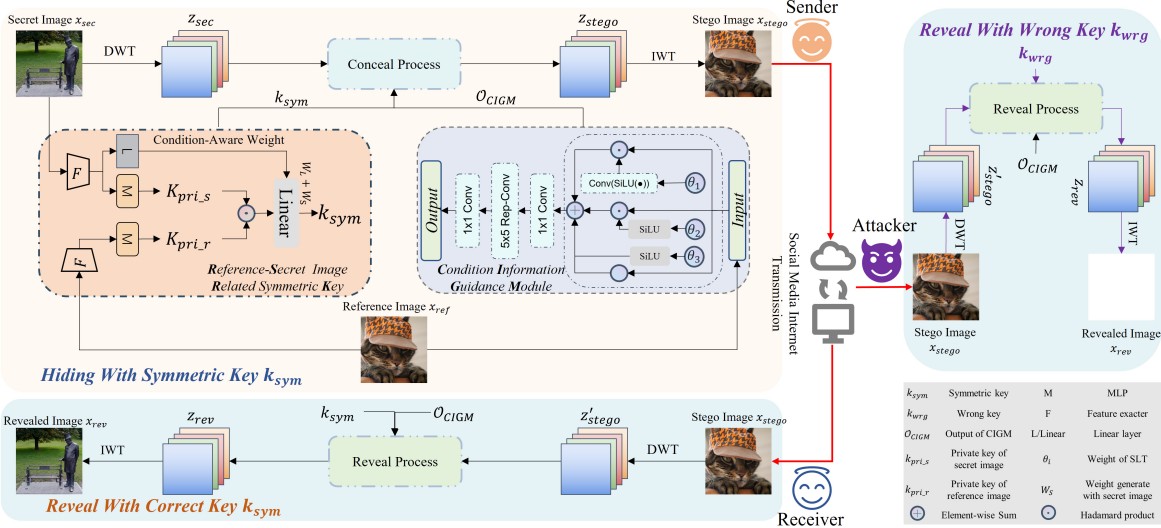

Figure 2. The overall structure of our SSHR model. In the conceal stage, the secret image $x_{sec}$ and reference image $x_{ref}$ jointly generate the symmetric key $k_{sym}$ first. Guided by $k_{sym}$ and $x_{ref}$, the secret image will be gradually encrypted and ultimately generate the stego image $x_{stego}$. In the reveal stage, both $k_{sym}$ and $x_{ref}$ are fed into the model simultaneously to accurately reveal the secret image $x_{sec}$.

Nevertheless, the limited control over text prompts, coupled with inconsistency between DDIM and its inversion, significantly damage the quality of stego and revealed secret images. This drives us to replace the text prompts with images and construct the exact reveal process. Furthermore, adaptive keys compensate for the absence of keys after removing text prompts.

### 2.3. Perona-Malik model

The Perona-Malik (PM) model (Perona & Malik, 1990) which has demonstrated strong performance in various image processing tasks, is a nonlinear diffusion model and formulated by the following partial differential equation:

$$\begin{cases} \frac{\partial z}{\partial t} = div(g(|\nabla z|)\nabla z), \\ z|_{t=0} = f, \end{cases} \quad (1)$$

where $\nabla$ represents the gradient operator, $t$ denotes the time, $f$ represents the initial image and $g$ is the diffusion function (Weickert et al., 1998). The discrete version of the PM model is derived through an explicit finite difference scheme and represented as:

$$\begin{aligned} \frac{z_{t+1} - z_t}{\Delta t} &= - \sum_{i \in x,y} \nabla_i^T \Lambda(z_t) \nabla_i z_t \\ &= - \sum_{i \in x,y} \nabla_i^T \phi(\nabla_i z_t), \end{aligned} \quad (2)$$

where $\phi$ is the influence function (Black et al., 1997) or flux function (Weickert et al., 1998). As demonstrated in previous works (Scherzer & Weickert, 2000; Zhu & Mumford,

1997), the diffusion step Equation (2) aligns with a gradient descent step to minimize the following energy function:

$$\mathcal{P}(z) = \sum_{i \in x,y} \sum_{p=1}^{N} \rho((k_i * z)_p), \quad (3)$$

where the functions $\rho$ is the penalty function, $k_i$ denotes a two-dimensional convolution filter kernel and $*$ signifies the convolution operation. It is noteworthy that the matrix-vector product $\nabla_x(z)$ can be interpreted as a 2D convolution of $z$ with the linear filter $k_x = [-1, 1]^T$. Similarly, $\nabla_y \in \mathcal{R}^{N \times N}$ corresponds to the linear filter $k_y = [-1, 1]^T$.

This paper systematically presents a novel and secure generative steganography model. Motivated by the limited control of text prompts, this paper introduces a secondary image with a new identity, replacing the text prompt for a more intuitive and effective approach. The exact reveal process is conducted to mitigate the inconsistency between the conceal and reveal process in existing methods. Moreover, the absence of keys after removing text prompts will be compensated with the adaptively generated symmetric keys. These keys will be integrated with reference images, serving as guiding elements throughout the process and enhancing security beyond conventional text prompts.

## 3. Proposed Methods

### 3.1. Framework

This paper proposes SSHR, a novel and secure generative steganography model, with the overall framework presented

in Figure 2. Building on the exceptional performance of prior work (Jing et al., 2021), our steganography process is conducted in the frequency domain. Specifically, the secret image $x_{sec}$ is first accepted as input and transformed into the latent space as $z_{sec}$ using discrete wavelet transform (DWT). Subsequently, the $z_{sec}$ is encrypted with the **symmetric key $k_{sym}$** and **the reference image $x_{ref}$**, which is pre-processed with the condition information guidance module (CIGM). This step generates the latent representation $z_{stego}$ of the stego image $x_{stego}$. The final stego image $x_{stego}$ is obtained through inverse wavelet transform (IWT). The reveal process follows the **exact inverse** of the conceal process, ultimately yielding the revealed secret image $x_{rev}$.

**Conceal process.** To meet the specific requirements of our steganography task, we introduce a condition term $\mathcal{R}(z, k_{sym}, c)$ into the original PM diffusion model. Referred to as the condition reaction term following the terminology in (Chen & Pock, 2016), this term serves to guide the generation process. Consequently, Equation (3) is modified as follows:

$$\mathcal{F} = \sum_{i=1}^{N_k} \rho_i(k_i * z, k_{sym}) + \lambda \mathcal{R}(z, k_{sym}, c), \quad (4)$$

where $k_{sym}$ denotes the Reference-Secret Image Related Symmetric Key, which is utilized to strengthen the security of both the key and the concealed secret images, with additional details provided later. $N_k$ represents the amount of the filters. $c$ is the condition infused into the condition reaction term to regulate the generated stego images and enhance their naturalness and imperceptibility. In our model, $c$ is set to the reference images by default. Equation (4) results in a Nonlinear Diffusion Model incorporating the condition reaction term, depicted as a forward-backward step at $z_{t-1}$ of the energy functional, expressed as:

$$E^t(x) = \sum_{i=1}^{N_k} \mathcal{P}_i^t(z, k_{sym}) + \mathcal{R}^t(z, k_{sym}, c), \quad (5)$$

where $\mathcal{P}_i^t(z, k_{sym}) = \sum_{j=1}^{N_k} \rho_i^t((K_i^t z)_j, k_{sym})$. By applying an explicit finite difference scheme, we derive the following discretized iteration law, with the initial input set as $z_0 = z_{sec}$:

$$z_{t+1} = z_t - \sum_{i=1}^{N_k} \bar{k}_i^t * \phi_i^t(k_i^t * z_t, k_{sym}) \\ - \lambda \varphi_t(z_t, k_{sym}, c). \quad (6)$$

To perform the convolution $k_i^t * z_t$, we substitute traditional convolution with Re-parameterization Depthwise Convolution (Rep-DWConv) (Tu et al., 2024), which

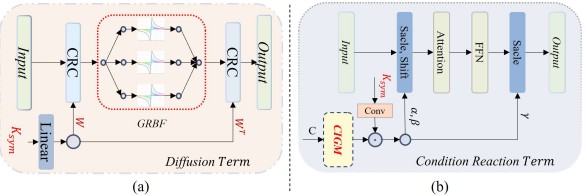

*Figure 3.* Illustration of (a) the Diffusion Term (DT) and (b) the Condition Reaction Term (CRT).

offers enhanced performance and parameter efficiency. The weights of Rep-DWConv, given by $W_{re-c} = Linear(SiLU(Linear(k_{sym})))$, are dynamically generated using the symmetric key $k_{sym}$ to ensure security. Consequently, this module is referred to as Conditional Re-parameterization Convolution (CRC). In line with (Chen & Pock, 2016), we use 63 Gaussian radial basis functions(GRBFs) to parameterize $\phi_i^t$ for $i \in \{1, 2, ..., N_k\}$. The overall pipeline for the Diffusion Term (DT) and the Condition Reaction Term (CRT) is illustrated in Figure 3.

Inspired by the coupling function and affine coupling layer introduced in (Dinh et al., 2014; 2016), we define an auxiliary variable $y_t$ that satisfies $y_t = z_t$. Consequently, the iteration step in Equation (6) is transformed into:

$$\begin{cases} y_t = z_t, \\ z_{t+1} = z_t \quad - \sum_{i=1}^{N_k} \bar{k}_i^t * \phi_i^t(k_i^t * y_t, k_{sym}) \\ \qquad\qquad - \lambda \varphi_t(y_t, k_{sym}, c). \end{cases} \quad (7)$$

Therefore, $y_{t+1}$ is calculated as:

$$\begin{aligned} y_{t+1} &= z_{t+1} \\ &= z_t - \sum_{i=1}^{N_k} \bar{k}_i^t * \phi_i^t(k_i^t * y_t, k_{sym}) \\ &\quad - \lambda \varphi_t(y_t, k_{sym}, c) \\ &= y_t - \sum_{i=1}^{N_k} \bar{k}_i^t * \phi_i^t(k_i^t * z_t, k_{sym}) \\ &\quad - \lambda \varphi_t(z_t, k_{sym}, c). \end{aligned} \quad (8)$$

By combining the equation above with the second equation in Equation (7), we obtain:

$$\begin{cases} z_{t+1} = z_t \quad - \sum_{i=1}^{N_k} \bar{k}_i^t * \phi_i^t(k_i^t * y_t, k_{sym}) \\ \qquad\qquad - \lambda \varphi_t(y_t, k_{sym}, c), \\ y_{t+1} = y_t \quad - \sum_{i=1}^{N_k} \bar{k}_i^t * \phi_i^t(k_i^t * z_t, k_{sym}) \\ \qquad\qquad - \lambda \varphi_t(z_t, k_{sym}, c). \end{cases} \quad (9)$$

To accelerate model convergence and drawing inspiration from the Gauss-Seidel method, we use $z_{t+1}$ instead of $z_t$ as

the input of the second equation, leading to:

$$
\begin{cases}
z_{t+1} = z_t & -\sum_{i=1}^{N_k} \bar{k}_i^t * \phi_i^t(k_i^t * y_t, k_{sym}) \\
& -\lambda \varphi_t(y_t, k_{sym}, c), \\
y_{t+1} = y_t & -\sum_{i=1}^{N_k} \bar{k}_i^t * \phi_i^t(k_i^t * z_{t+1}, k_{sym}) \\
& -\lambda \varphi_t(z_{t+1}, k_{sym}, c).
\end{cases}
\tag{10}
$$

Following the recommendation in (Wallace et al., 2023), we introduce a learnable parameter $p \in [0.90, 1.0]$ to enhance robustness. Additionally, intermediate mixing layers are applied after each gradient descent step to compute weighted averages of $z$ and $y$, defined as:

$$
z^{mix} = p \cdot z + (1 - p) \cdot y.
\tag{11}
$$

In summary, the conceal process of the proposed SSHR, with the input $z_0 = z_{sec}$ and final output $z_T = z_{stego}$, can be formulated as:

$$
\begin{cases}
z_{t+1}^{mix} = z_t - \sum_{i=1}^{N_k} \bar{k}_i^t * \phi_i^t(k_i^t * y_t, k_{sym}) \\
\qquad - \lambda \varphi_t(y_t, k_{sym}, c), \\
y_{t+1}^{mix} = y_t - \sum_{i=1}^{N_k} \bar{k}_i^t * \phi_i^t(k_i^t * z_{t+1}^{mix}, k_{sym}) \\
\qquad - \lambda \varphi_t(z_{t+1}^{mix}, k_{sym}, c), \\
z_{t+1} = p \cdot z_{t+1}^{mix} + (1 - p) \cdot y_{t+1}^{mix}, \\
y_{t+1} = p \cdot y_{t+1}^{mix} + (1 - p) \cdot z_{t+1}.
\end{cases}
\tag{12}
$$

**Exact reveal process.** To minimize errors in the reveal process and enhance the quality of the revealed secret images, we deterministically reverse the conceal process, ensuring a reliable reveal process in the proposed model. Equation (11) represents an invertible affine transformation, with the inversion process defined as follows:

$$
z = (z^{mix} - (1 - p) \cdot y)/p.
\tag{13}
$$

Thus, reveal Equation (12) and the exact reveal process of our model is calculated, with input $z_T = z_{stego}$ and output $z_0 = z_{rev}$, as follows:

$$
\begin{cases}
y_{t+1}^{mix} = (y_{t+1} - (1 - p) \cdot z_{t+1})/p, \\
z_{t+1}^{mix} = (z_{t+1} - (1 - p) \cdot y_{t+1}^{mix})/p, \\
y_t = y_{t+1}^{mix} + \sum_{i=1}^{N_k} \bar{k}_i^t * \phi_i^t(k_i^t * z_{t+1}^{mix}, k_{sym}) \\
\qquad + \lambda \varphi_t(z_{t+1}^{mix}, k_{sym}, c), \\
z_t = z_{t+1}^{mix} + \sum_{i=1}^{N_k} \bar{k}_i^t * \phi_i^t(k_i^t * y_t, k_{sym}) \\
\qquad + \lambda \varphi_t(y_t, k_{sym}, c).
\end{cases}
\tag{14}
$$

In practice, the calculation steps of $z$ and $y$ in both the conceal and reveal process are executed alternately, ensuring a symmetrical transformation between the two sequences.

### 3.2. Reference-Secret Image Related Key

Ensuring the security of the secret image hidden within the stego images is of paramount in image steganography. The concept of the key, originally from cryptography and first introduced to steganography in (Mou et al., 2023), is well-suited to address this requirement. However, existing methods do not fully guarantee key security and vulnerable to key guessing. In (Mou et al., 2023), the key is merely a single numeric value; in CRoSS (Yu et al., 2024), the private and public keys correspond to the descriptive text of the secret and stego images, respectively. Similarly, DiffStega (Yang et al., 2024) employs a pre-determined password-related reference image, alongside a text prompt as the public key. In contrast, and drawing inspiration from the Elliptic Curve Diffie-Hellman Ephemeral (ECDHE) key exchange algorithm, our model introduces a dynamically generated symmetric key that is adaptively derived from the reference-secret image pair, significantly boosting security.

Inspired by the principle of ECDHE, we treat each steganography task as an independent encryption session, where the symmetric key is adaptively generated based on the secret and reference images. The process begins by extracting features $F_s$ and $F_r$ from the secret and reference image, respectively, using a pre-trained neural network, with AlexNet employed as the default model. Next, a MLP is applied to generate the private key, defined as $k_{pri-i} = MLP(F_i), i \in r, s$, where $k_{pri-r}$ and $k_{pri-s}$ represent the private keys of the reference and secret images, respectively. The public key is then derived using a specific parameter matrix $W$, formulated as $k_{pub-i} = W \cdot k_{pri-i}$, where $\cdot$ represents the Hadamard product.

Following the principles of the ECDHE algorithm, the symmetric key $k_{sym}$ utilized in our model is derived as:

$$
\begin{aligned}
k_{sym} &= (W_L + W_S) \cdot k_{pri-s} \cdot k_{pri-r} \\
&= (W_L + W_S) \cdot k_{pri-r} \cdot k_{pri-s},
\end{aligned}
\tag{15}
$$

where $W_S$ denotes the weight matrix generated based on the secret image, and the overall framework is illustrated in Figure 2. Based on the above equation, the parameter matrix $W$ used for public key generation is defined as $W = W_L + W_S$. Thus, for each unique pair of images, a distinct symmetric key is generated and used in both the conceal and reveal processes. During transmission, only the public key is shared, ensuring strict protection of both the private keys and the symmetric key. This approach significantly enhances the security of the secret images concealed within the stego images.

When transmitting the stego image $x_{stego}$, the public key

$k_{pub-s}$ associated with the secret image is sent alongside it. Upon receiving the public key of the secret image, the receiver can use the same pipeline outlined in Equation (15) to generate the symmetric key $k_{sym}$, as the sender did during the conceal process. With this symmetric key, the receiver can accurately reveal the secret image and complete the reveal process. Further details are provided in the Supplementary. To the best of our knowledge, this is the first exploration to integrate key exchange protocols into image steganography.

Subsequently, the symmetric key $k_{sym}$ is utilized to generate the weights for CRC, denoted as $W_{re-c}$, as well as the modulation parameters $\alpha$, $\beta$, $\gamma$ in CRT. This design ensures that the symmetric key plays a crucial role in the conceal and reveal process.

### 3.3. Condition Information Guidance Module

We aim to progressively encrypt the secret images, allowing the stego images to gradually improve in quality and increasingly resemble the reference images over time. To achieve this, we first pre-process the reference images using the Condition Information Guidance Module (CIGM) before generating the modulation parameters. This ensures that the input to AdaLN-Zero (Peebles & Xie, 2023) varies across different timesteps, thereby enabling the stego images to incrementally converge toward the reference images. The CIGM is analogous to a transformer block and comprises a a separate linear transformation(SLT) (Lu et al., 2024), followed by a Re-parameterization Feed-Forward Network(Re-FFN)(Tu et al., 2024). The SLT serves a function similar to that of the attention module in a transformer block and can be formulated as follows:

$$\mathcal{O}_{slt}(c) = W(\theta_1) \cdot c + \theta_2 \cdot c + \theta_3 \,, \quad (16)$$

where $\theta_1 \in \mathbf{R}^{B \times C \times H \times W}$, $\theta_2 \in \mathbf{R}^{B \times C \times 1 \times 1}$, $\theta_3 \in \mathbf{R}^{B \times C \times 1 \times 1}$ and $c$ is the input of it. The $W(\theta_1)$ is a convolution layer, expressed as $W(\theta_1) = Conv(SiLU(\theta_1))$. The Re-FFN is structured with two pointwise convolution layers and a Rep-Conv layer, as illustrated in Figure 2.

Subsequently, the modulation parameters $\alpha$, $\beta$, $\gamma$ are generated based on the output of CIGM, denoted as $\mathcal{O}_{CIPM}(c)$, along with the parameter matrix $W_A$. The parameters $W_A$ derived from the symmetric key $k_{sym}$ via a MLP layer. The overall pipeline is formulated as:

$$\begin{aligned} \alpha, \beta, \gamma &= split((\mathcal{O}_{CIGM}(c)) \cdot W_A) \\ &= split((\mathcal{O}_{CIGM}(c)) \cdot MLP(k_{sym}) \,. \end{aligned} \quad (17)$$

Subsequently, they are incorporated into the CRT to guide the generation process and encrypt the secret image.

In summary, the output of CRT $\mathcal{O}_{CRT}(z, k_{sym}, c)$ is:

$$\mathcal{O}_{CRT}(z, k_{sym}, c) = \gamma \cdot N(z \cdot (1 + \alpha) + \beta), \quad (18)$$

where $N$ is the Transformer block illustrated in Figure 3.

### 3.4. Loss function

Our loss function comprises frequency loss and perceptual loss, which we will describe in detail below.

**Frequency loss.** The steganography process intends to generate the stego image $x_{stego}$ based on the secret image $x_{sec}$. For security purposes, the frequency content of the stego image $z_{stego}$ should closely match that of the reference image $z_{ref}$, making them indistinguishable. In the reveal phase, it is essential that the frequency profile of the revealed image $z_{rev}$ closely aligns with that of the original secret image $z_{sec}$. To enforce these constraints, we define the frequency loss as follows:

$$\mathcal{L}_F = l_s(z_{ref}, z_{stego}) + l_s(z_{sec}, z_{rev}), \quad (19)$$

where $l_s$ represents the $l_1$ or $l_2$ norm, serving as a measure of the difference between two images. In our experiments, we use the $l_1$ norm as the default.

**Perceptual loss.** To ensure that the stego image $x_{stego}$, generated during the conceal process, closely resembles the reference image $x_{ref}$, and that the revealed image $x_{rev}$ visually aligns with the secret image $x_{sec}$, we introduce a perceptual loss. This loss function quantifies the high-level perceptual differences between these image pairs. The perceptual loss is defined as follows:

$$\begin{aligned} \mathcal{L}_P = &l_p(\mathscr{A}(x_{ref}), \mathscr{A}(x_{stego})) \\ &+ l_p(\mathscr{A}(x_{sec}), \mathscr{A}(x_{rev})), \end{aligned} \quad (20)$$

where $\mathscr{A}$ is the pre-trained AlexNet (Krizhevsky et al., 2012), used to extract high-level features from the images, and $l_p$ measures the difference between these features.

**Total loss.** The total loss function $\mathcal{L}_{Total}$ is defined as the weighted sum of the frequency loss $\mathcal{L}_F$ and perceptual loss $\mathcal{L}_P$, formulated as:

$$\mathcal{L}_{Total} = \lambda_1 \mathcal{L}_F + \lambda_2 \mathcal{L}_P, \quad (21)$$

where $\lambda_1$ and $\lambda_2$ are trade-off parameters set to 2.0 and 1.0, respectively, to balance the different losses.

## 4. Experiments

### 4.1. Experimental Setting

**Datasets and settings.** Our model is implemented with PyTorch and trained on the DIV2K (Agustsson & Timofte, 2017) training dataset. The evaluation is performed on the DIV2K (Agustsson & Timofte, 2017) test dataset (100 images), COCO (Lin et al., 2014) (5000 images), ImageNet (Russakovsky et al., 2015) (10,000 images), and UniStega (Yang et al., 2024) (100 images) at a resolution of $256 \times 256$. Additional details are presented in the Supplementary.

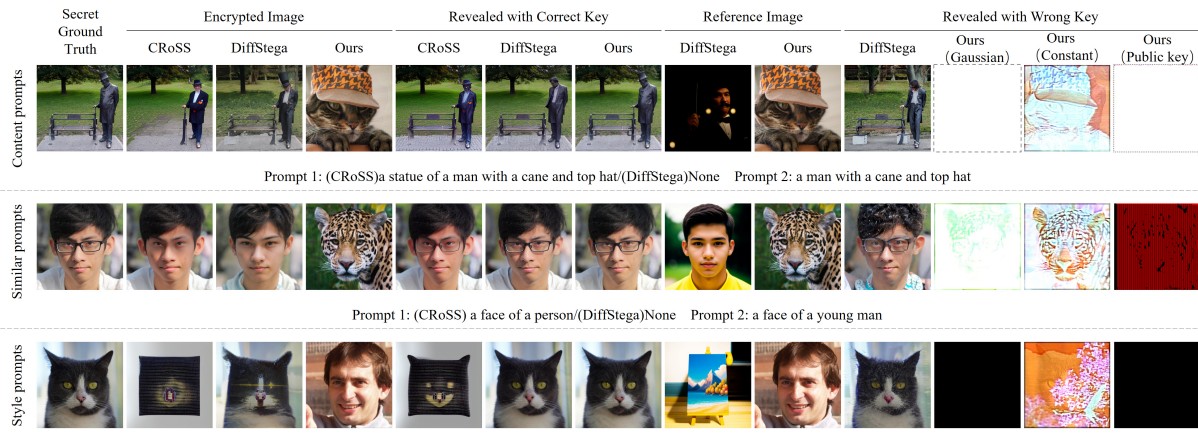

*Figure 4.* Visual comparisons of our model with generative steganography models on the UniStega dataset across three different prompts. These prompts are utilized in CRoSS and DiffStega, whereas our model functions without text prompts. The reference image is used as the image condition in both DiffStega and our model. More comparison results are presented in the Supplementary.

*Table 1.* Numerical comparisons of secret/revealed image pairs with cover-based steganography methods across various datasets, highlighting the best results in red and the second-best in bold.

| METHOD | DIV2K | | | | COCO | | | | IMAGENET | | | |
|---|---|---|---|---|---|---|---|---|---|---|---|---|
| | PSNR↑ | SSIM↑ | MAE↓ | RMSE↓ | PSNR↑ | SSIM↑ | MAE↓ | RMSE↓ | PSNR↑ | SSIM↑ | MAE↓ | RMSE↓ |
| HiDDeN | 36.43 | 0.9496 | 6.02 | 5.50 | 37.68 | 0.9145 | 4.72 | 6.33 | 35.70 | 0.9301 | 4.57 | 6.92 |
| Baluja | 35.88 | 0.9377 | 4.68 | 6.11 | 35.01 | 0.9341 | 6.52 | 8.00 | 34.13 | 0.9247 | 5.31 | 8.37 |
| Weng et.al | 33.24 | 0.8582 | 5.04 | 6.32 | 33.05 | 0.8921 | 4.80 | 6.06 | 33.34 | 0.8921 | 4.80 | 6.06 |
| UDH | 35.22 | 0.8036 | 4.03 | 4.78 | 35.07 | 0.8220 | 3.77 | 4.67 | 35.39 | 0.8252 | 3.73 | 4.58 |
| ISN | 37.06 | 0.9672 | 2.80 | 4.30 | 36.58 | 0.9016 | 3.04 | 3.78 | 37.73 | 0.9548 | 2.97 | 3.31 |
| HiNet | **46.64** | **0.9962** | **0.93** | **1.31** | **44.05** | **0.9952** | **1.17** | **1.70** | **46.78** | **0.9947** | **1.12** | **1.23** |
| OURS | 48.56(1.92↑) | 0.9988(0.0022↑) | 0.74(0.19↓) | 0.97(0.34↓) | 47.67(3.62↑) | 0.9985(0.0033↑) | 0.80(0.37↓) | 1.08(0.62↓) | 49.52(2.74↑) | 0.9986(0.0039↑) | 0.68(0.44↓) | 0.88(0.35↓) |

*Table 2.* Numerical comparisons with generative steganography methods on the UniStega dataset.

| METHOD | STEGO IMAGES | | CORRECT KEY | | WRONG KEY(CONSTANT) | |
|---|---|---|---|---|---|---|
| | PSNR↓ | SSIM↓ | PSNR↑ | SSIM↑ | PSNR↓ | SSIM↓ |
| CRoSS | 19.03 | 0.66 | 21.25 | 0.71 | - | - |
| DiffStega | **18.61** | **0.59** | 23.29 | 0.77 | 17.53 | 0.54 |
| DiffStega‡ | 19.73 | 0.65 | **23.92** | **0.79** | 20.68 | 0.70 |
| OURS | 8.96(9.65↓) | 0.11(0.48↓) | 47.04(23.12↑) | 0.99(0.20↑) | 5.70(11.83↓) | 0.24(0.30↓) |

*Table 3.* NIQE scores for various models.

| | ORIGINAL IMAGE | ISN | HiNet | CRoSS | DiffStega | OURS |
|---|---|---|---|---|---|---|
| NIQE↓ | 5.14 | 11.28 | **5.35** | 5.60 | 5.58 | **5.46** |

tural Similarity Index Measure (SSIM) (Wang et al., 2004), Root Mean Square Error (RMSE), and Mean Absolute Error (MAE) as performance metrics. In addition, we use the Naturalness Image Quality Evaluator (NIQE) (Mittal et al., 2012) to evaluate the naturalness of the stego images.

### 4.2. Quality Analysis

**Quantitative results.** We compare the proposed method against existing models, with numerical results presented in Table 1 and Table 2. Table 1 presents a quantitative comparison between the proposed SSHR model and other cover-based models on the DIV2K, COCO, and ImageNet datasets. Notably, on the ImageNet dataset, our model demonstrates a PSNR gain of **2.74** dB and an SSIM improvement of **0.39%**. Meanwhile, MAE and RMSE are reduced by **0.44** and **0.35**, respectively. Similarly, on the DIV2K and COCO datasets, the proposed SSHR model outperforms other cover-based models, achieving PSNR/SSIM gains of **1.92** dB/**0.22%** and **3.62** dB/**0.33%**, respectively, along with lower MAE and RMSE. These results illustrate that SSHR significantly enhances the quality of revealed secret images compared to existing cover-based steganography methods.

Table 2 presents the quantitative results in comparison with generative methods on the UniStega dataset. Specifically,

**Benchmarks.** To comprehensively evaluate our method's effectiveness, we compare it against state-of-the-art (SOTA) steganography methods, including cover-based methods: Baluja et al. (Baluja, 2017), HiDDeN (Zhu, 2018), UDH (Zhang et al., 2020), Weng et al. (Weng et al., 2019), ISN (Lu et al., 2021), HiNet (Jing et al., 2021); and generative methods: CRoSS (Yu et al., 2024), DiffStega (Yang et al., 2024). To maintain objectivity, we re-trained the cover-based models with the same training dataset as ours.

**Evaluation Metrics.** To assess the quality of secret/revealed pairs, we utilize Peak Signal-to-Noise Ratio (PSNR), Struc-

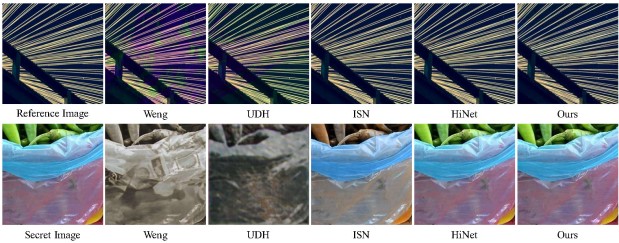

*Figure 5.* Visual comparisons of stego images(top) and revealed secret images(bottom) for our model and various cover-based steganography models on the DIV2K dataset.

our model achieves an impressive **24** dB gain in PSNR and **20%** improvement in SSIM for correct recovery/secret pairs, indicating a substantial enhancement. For secret/stego image pairs, the proposed model reduces PSNR by **9.65** dB and SSIM by **48%** compared to other generative methods, effectively diminishing the resemblance between stego and their corresponding secret images. Additionally, lower similarity scores for incorrect recovery/secret image pairs, with a notable **11.83 dB** decrease in PSNR and **30%** reduction in SSIM, demonstrate the proposed model effectively bolsters security for recovery attempts with incorrect key. Evidently, the SSHR excels in encryption and decryption compared to other generative steganography models, offering high-quality revealed secret images and enhanced security.

**Qualitative Results.** The qualitative comparison outcomes for the stego and recovery images of our model and other models are presented in Figure 4 and Figure 5. Figure 4 illustrates comparisons between our SSHR model and other generative steganography models on the UniStega dataset. The results demonstrate that the proposed model generates stego images that are distinctly different from the secret images, while simultaneously producing higher-quality revealed secret images. Additionally, the SSHR model shows enhanced security when the image is revealed with three types of incorrect keys. Figure 5 showcases the visual performance of our model in comparison to various cover-based steganography methods. The figure highlights the superior performance of the proposed method in terms of secret/reveal image pairs, as well as the comparative results for cover/stego image pairs. These results clearly demonstrate that the proposed model achieves significant improvements in effectiveness and security over existing SOTA models.

### 4.3. Security Analysis

**Naturalness and Imperceptibility.** We apply the NIQE to evaluate the naturalness and visual security of the images against human suspicion without the assistance of reference images or human feedback. The results in Table 3 show a 0.12 reduction in NIQE value for SSHR model compared to other generative steganography models. This indicates that

*Table 4.* The detection accuracy (%) detected by SRNet and XuNet.

| | COVER-BASED METHOD | | | | GENERATIVE METHOD | | |
|---|---|---|---|---|---|---|---|
| | WENG | UDH | ISN | HINET | CROSS | DIFFSTEGA | OURS |
| SRM | 81.47 | 76.34 | 54.86 | 53.52 | 51.73 | 51.68 | **50.64(1.04↓)** |
| SRNET | 85.31 | 79.82 | 55.68 | 55.54 | 52.10 | 52.03 | **51.05(0.98↓)** |
| XUNET | 86.24 | 80.26 | 55.42 | 55.37 | 53.12 | 52.86 | **50.93(1.93↓)** |

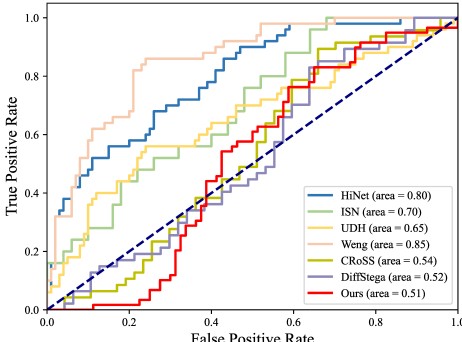

*Figure 6.* Security performance detected by StegExpose.

the proposed model outperforms other generative steganography models in terms of naturalness and imperceptibility.

**Steganographic analysis.** The anti-steganalysis ability is crucial for evaluating the security of image steganography, as it measures the likelihood that stego images can be distinguished from reference images using steganalysis tools. Following the common practices in the cover-based and generative steganography task (*e.g.*, ISN (Lu et al., 2021), HiNet (Jing et al., 2021), CRoSS (Yu et al., 2024), DiffStega (Yang et al., 2024)), we employ the open-source steganalysis tool StegExpose (Boehm, 2014), hand-crafted feature-based steganalyzer SRM (Fridrich & Kodovsky, 2012) and two steganalysis networks: SRNet (Boroumand et al., 2018) and XuNet (Xu et al., 2016), to systematically evaluate the anti-steganalysis capabilities of the proposed model alongside other methods. Lower detection accuracy and a smaller area under curve (AUC) indicates better security performance. The evaluate results are presented in Figure 6 and Table 4 respectively. These steganalysis results indicate that the proposed SSHR model outperforms other SOTA methods in terms of anti-steganalysis performance.

**Effect of the Symmetric Key.** The RSRK is designed to secure both the key and the concealed secret image. We tested the model with the correct symmetric key, alongside three types of incorrect keys: random Gaussian noise, a constant value, and the public key of the secret image (which could be intercepted by attackers during transmission). As shown in Figure 1 and Figure 4, only the correct key enables accurate recovery of the secret image. Figure 1 uses a constant key as the incorrect key, and Table 2 provides numerical results for revealing the secret image with a constant key. Both numerical and visual results consistently show that an

*Table 5.* Effectiveness of PM, Wavelet transform,Rep-Conv and CIGM.

| PM | WAVELET TRANSFORM | REP-CONV | CIGM | REFERENCE/STEGO | | | | SRCRET/RECOVERY | | | |
|---|---|---|---|---|---|---|---|---|---|---|---|
| | | | | PSNR | SSIM | MAE | RMSE | PSNR | SSIM | MAE | RMSE |
| ✗ | ✗ | ✗ | ✗ | 30.25 | 0.7862 | 8.57 | 9.23 | 37.77 | 0.9489 | 3.58 | 4.23 |
| ✓ | ✗ | ✗ | ✗ | 35.37 | 0.8050 | 7.77 | 7.90 | 41.36 | 0.9624 | 2.75 | 2.88 |
| ✓ | ✓ | ✗ | ✗ | 37.96 | 0.9318 | 2.99 | 3.57 | 42.24 | 0.9969 | 1.69 | 2.18 |
| ✓ | ✓ | ✓ | ✗ | 39.92 | 0.9670 | 2.16 | 2.91 | 45.19 | 0.9758 | 0.98 | 1.45 |
| ✓ | ✓ | ✓ | ✓ | **41.32** | **0.9897** | **1.13** | **1.25** | **48.56** | **0.9988** | **0.74** | **0.97** |

*Table 6.* Quantitative results of revealed images on the UniStega dataset when stego images undergo various distortion.

| METHOD | CLEAN | GAUSSIAN BLUR | GAUSSIAN NOISE | JPEG COMPRESSION | POISSON |
|---|---|---|---|---|---|
| ISN | 21.73 | 5.16 | 3.97 | 4.33 | 4.05 |
| HINET | 46.55 | 9.32 | 9.97 | 10.13 | 10.29 |
| CROSS | 21.25 | 20.08 | **18.97** | 20.20 | 19.27 |
| DIFFSTEGA | 23.29 | **20.85** | 18.62 | **21.16** | **20.15** |
| OURS | 47.04 | 24.01(3.16↑) | 23.95(4.98↑) | 24.74(3.58↑) | 23.96(3.81↑) |

attacker cannot retrieve the secret image with an incorrect key. These findings underscore the essential role of the proposed RSRK and key generation module in enhancing the security of both the key and the concealed secret image.

**Effectiveness of various setting.** As indicated in Table 5, the various settings we employed significantly improve the quantitative metrics for both stego and revealed images, enhancing performance in both the conceal and reveal processes. For example, the introduced PM model leads to a PSNR gain of 5.12 dB for reference/stego pairs and 3.59 dB for secret/recovery pairs. Figure 7 illustrates the encryption and decryption process, showcasing how the SSHR model achieves secure, stepwise encryption and decryption of secret images with reference image guidance. These results demonstrate that the proposed model effectively reduces the similarity between the stego and secret images, while producing high-quality revealed secret images, validating the module's effectiveness and overall model performance.

**Performances on Various Distortions.** To assess the robustness of our SSHR model, we test it under various degradations, including Gaussian blur, Gaussian noise, JPEG compression and Poisson noise. The results, presented in Table 6 in terms of PSNR, indicate that the proposed SSHR model outperforms other SOTA models. These results demonstrate the favorable robustness of the proposed SSHR against attacks. Additional details on the distortions are provided in the Supplementary.

## 5. Conclusion

This paper presents SSHR, an innovative steganography method based on the PM diffusion model, designed to overcome the limitations of existing generative steganography approaches. The proposed model uses reference images to guide stego image generation, ensuring outputs are visually natural yet highly dissimilar to the secret images, thereby enhancing concealment effectiveness. We also establish an

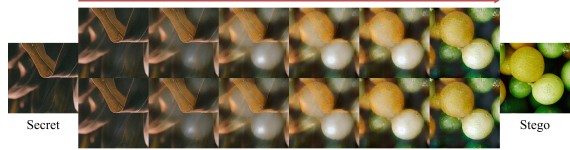

*Figure 7.* The CIGM facilitates the achievement of progressive encryption.

exact reveal process to improve the quality of the revealed secret images. Additionally, we propose an RSRK generation module that strengthens key security by dynamically linking them to both the reference and secret images, significantly enhancing the overall security of the concealed secret images. Extensive experiments demonstrate that SSHR outperforms SOTA generative steganography methods in both effectiveness and security.

## Acknowledgements

This work was supported in part by the NSFC fund (NO. 62206073, 62176077), in part by the Shenzhen Key Technical Project (NO. JCYJ20241202123728037, NO. JSGG20220831092805009, JSGG20220831105603006, JSGG20201103153802006, KJZD20230923115117033, KJZD20240903100712017), in part by the Guangdong International Science and Technology Cooperation Project (NO. 2023A0505050108), in part by the Shenzhen Fundamental Research Fund (NO. JCYJ20210324132210025), and in part by the Guangdong Provincial Key Laboratory of Novel Security Intelligence Technologies (NO. 2022B1212010005), in part by the Guangdong Shenzhen joint Youth Fund under Grant 2021A151511074, and in part by the Natural Science Foundation of Shenzhen General Project under Grant JCYJ20240813110007010, in part by the Natural Science Foundation of Guangdong Province under Grant 2023A1515010893, in part by the Shenzhen Doctoral Initiation Technology Plan under Grant RCBS20221008093222010, in part by the Shenzhen Pengcheng Peacock Startup Fund.

## Impact Statement

This paper delves into an exploration of generative image steganography technology, with the objective of generating the stego images from secret images. Through a succession of innovative algorithmic designs and meticulous experimental analyses, it has significantly boosted the security and effectiveness of generative image steganography methods. The integration of the key-exchange algorithm has effectively forged an initial connection between image steganography and cryptology. This linkage not only enriches the theoretical framework of image steganography

but also endows cryptology with new application scenarios.

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

# A. Security of the symmetric key

The ECDHE algorithm, widely used in the Transport Layer Security (TLS) handshake, is an effective key exchange method based on elliptic curves. Inspired by this, we treat each steganography task as an independent encryption session, adaptively generating symmetric keys linked to both the secret and reference images using the Reference-Secret Image Related Symmetric Key Generation (RSRK) module. This approach enhances the security of both the keys and the concealed secret images during the conceal and reveal processes.

We model each steganography task as an independent encryption session, where the secret image and reference image function as the client and server, respectively. Initially, features $F_s$ and $F_r$ are extracted from the secret and reference images using AlexNet. An Multilayer Perceptron (MLP) is then used to generate the private keys, formulated as $k_{pri-i} = MLP(F_i), i \in r, s$, where $k_{pri-r}$ and $k_{pri-s}$ are the private keys of the reference and secret images, respectively. These private keys are securely stored to maintain their integrity. Using specific parameters $W$, analogous to the base point $G$ in the ECDHE algorithm, we derive the public keys for the reference image $k_{pub-r}$ and secret image $k_{pub-s}$ as follows:

$$\begin{aligned} k_{pub-r} &= W \cdot k_{pri-r} \,, \\ k_{pub-s} &= W \cdot k_{pri-s} \,, \end{aligned} \tag{22}$$

where $\cdot$ denotes the Hadamard product, which is commutative and satisfies the necessary conditions for the subsequent operations in the ECDHE algorithm.

After obtaining the private and public key for the secret and reference image, we follow the principle of the ECDHE algorithm to derive the symmetric key $k_{sym}$ utilized in our model, expressed as follows:

$$\begin{aligned} k_{sym} &= k_{pub-s} \cdot k_{pri-r} \\ &= (W_L + W_S) \cdot k_{pri-s} \cdot k_{pri-r} \\ &= (W_L + W_S) \cdot k_{pri-r} \cdot k_{pri-s} \\ &= k_{pub-r} \cdot k_{pri-s} \,, \end{aligned} \tag{23}$$

where $W_S$ represents the weight generated based on the secret image, ensuring that the weight used to generate the symmetric key is dynamically tied to the secret image. This operation guarantees that different parameters are used to derive the public key $k_{pub-s}$ and the symmetric key $k_{sym}$ when concealing different secret images, analogous to selecting specific base points $G$ and elliptic curves $E$ in the ECDHE algorithm. This not only aligns the symmetric key generation process with the principles and procedures of the ECDHE algorithm but also enhances the security of the symmetric key. Following the above equation, the parameters $W$ are used for generating the public key and are defined as $W = W_L + W_S$. As a result, for each unique combination of images, a distinct symmetric key $k_{sym}$ is generated and used in both the conceal and reveal process.

In practical application, we assume that the sender and receiver share the same reference image and employ the same image preprocessing methods. Additionally, the model parameters used for generating the private key are common to both parties. Therefore, when using the same reference image, both the sender and receiver can derive the same private key $k_{pri-r}$ corresponding to the reference image.

When transmitting the stego image $x_{stego}$, the public key $k_{pub-s}$ associated with the secret image is sent alongside it. Upon receiving the public key of the secret image, the receiver can use the same pipeline outlined in Equation (23) to generate the symmetric key $k_{sym}$, as the sender did during the conceal process. With this symmetric key, the receiver can accurately reveal the secret image and complete the decrypt process.

The stego image $x_{stego}$ and public key $k_{pub-s}$ can be intercepted by an attacker during transmission. In existing generative steganography methods, the security of the keys is not guaranteed, exposing the secret image $x_{sec}$ to potential risk. In contrast, in our method, even if an attacker intercepts the stego image and the public key, they cannot obtain the correct symmetric key $k_{sym}$ as the private key $k_{pri-r}$ of the reference image $x_{ref}$ is never transmitted. Without the correct symmetric key, the attacker is incapable of revealing the secret image $x_{sec}$ from the stego image. This ensures the security of the key and significantly enhance the protection of the concealed secret image.

The proposed key generation process, grounded in robust, well-established algorithms, mitigates potential theoretical vulnerabilities. Specifically, the symmetric key generation method employs the Elliptic Curve Diffie-Hellman Ephemeral (ECDHE) algorithm. The security of key exchange protocols (e.g. RSA, ECDH) is mathematically grounded in computationally hard

problems like integer factorization for RSA, and elliptic curve discrete logarithm problem for ECDH and so on. These foundational problems guarantee the computational infeasibility of deriving private keys or shared secrets from publicly available parameters. This ensures strong theoretical security guarantees, safeguarding the symmetric key's generation and protection.

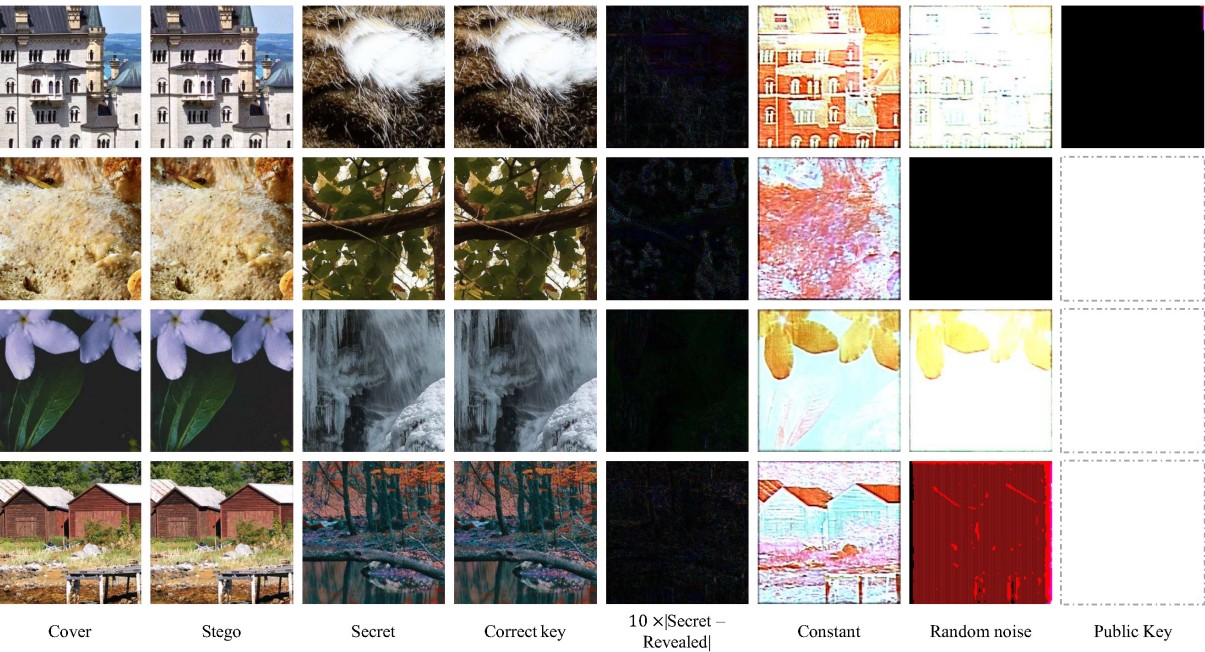

| Cover | Stego | Secret | Correct key | 10 ×\|Secret − Revealed\| | Constant | Random noise | Public Key |

*Figure 8.* Visual results on DIV2K dataset. The secret images are revealed with four types of keys: correct keys, constant, random noise and the public keys tied to secret images.

## B. Implementation Details

**Experimental Setting.** The model is implemented in PyTorch and trained on the DIV2K (Agustsson & Timofte, 2017) training dataset. Training images are randomly cropped to $256 \times 256$ and augmented with random horizontal and vertical flips. Evaluation is performed on DIV2K (Agustsson & Timofte, 2017) (100 images), COCO (Lin et al., 2014) (5000 images), ImageNet (Russakovsky et al., 2015) (10,000 images), and UniStega (Yang et al., 2024) (100 images). Comparatively, images in the DIV2K dataset are center-cropped, while in the other datasets, the images are resized to $256 \times 256$. The AdamW optimizer with an initial learning rate of $1 \times 10^{-4}$ is used for training. All experiments are conducted on a Nvidia 4090 GPU.

**Degradation setting.** To evaluate the robustness of our SSHR model, we test it under a variety of degradations, including Gaussian blur, Gaussian noise, JPEG compression, and Poisson noise. For Gaussian blur, we apply a Gaussian kernel with $\sigma = 0.1 \times std$ to blur the stego images via convolution with a kernel size of 3, where $std$ denotes the standard deviation of the stego images. In the case of Gaussian noise, we generate the noise with $\sigma = 0.1 \times std$. In addition, we set the quality factor $Q = 75$ for JPEG compression in our experiments.

## C. Additional Results

Table 7 provides a numerical comparison of the reference/stego image pairs with cover-based steganography methods across the DIV2K, COCO, and ImageNet datasets. For instance, on the ImageNet dataset, our model achieves a PSNR improvement of **1.81** dB and an SSIM enhancement of **0.64%**. Additionally, both MAE and RMSE are reduced by **0.37** and **0.35**, respectively. Similarly, on the DIV2K and COCO datasets, the SSHR model outperforms other cover-based methods,

*Table 7.* Numerical comparisons of reference/stego image pairs with cover-based steganography methods across various datasets, highlighting the best results in red and the second-best in bold.

| Method | Reference/Stego | | | | | | | | | | | |
|---|---|---|---|---|---|---|---|---|---|---|---|---|
| | DIV2K | | | | COCO | | | | Imagenet | | | |
| | PSNR↑ | SSIM↑ | MAE↓ | RMSE↓ | PSNR↑ | SSIM↑ | MAE↓ | RMSE↓ | PSNR↑ | SSIM↑ | MAE↓ | RMSE↓ |
| HiDDeN | 35.21 | 0.9691 | 6.98 | 6.82 | 36.71 | 0.9676 | 6.58 | 8.73 | 34.79 | 0.9380 | 6.12 | 7.33 |
| Baluja | 36.77 | 0.9645 | 3.79 | 5.02 | 36.38 | 0.9563 | 5.98 | 7.43 | 36.59 | 0.9520 | 5.61 | 5.41 |
| Weng et.al | 37.34 | 0.9341 | 3.03 | 3.57 | 37.68 | 0.9323 | 2.82 | 3.42 | 38.20 | 0.9368 | 2.68 | 3.24 |
| UDH | 38.78 | 0.9658 | 2.82 | 2.94 | 38.90 | 0.9650 | 2.77 | 2.90 | 38.96 | 0.9624 | 2.75 | 2.88 |
| ISN | 39.28 | 0.9853 | 2.34 | 2.91 | 37.95 | 0.9751 | 2.76 | 3.23 | 40.13 | 0.9748 | 1.95 | 2.51 |
| HiNet | **39.53** | **0.9868** | **2.08** | **2.87** | **39.01** | **0.9844** | **2.09** | **2.96** | 44.61 | 0.9927 | 1.52 | 1.63 |
| Ours | 41.03(1.50↑) | 0.9894(0.0026↑) | 1.65(0.43↓) | 1.93(0.94↓) | 40.21(1.20↑) | 0.9885(0.0041↑) | 1.63(0.46↓) | 1.90(1.06↓) | 46.42(1.81↑) | 0.9991(0.0064↑) | 1.15(0.37↓) | 1.28(0.35↓) |

with PSNR/SSIM improvements of **1.50** dB/**0.26%** and **1.20** dB/**0.41%**, respectively, alongside lower MAE and RMSE values. These results demonstrate that the SSHR model substantially enhances the quality and imperceptibility of stego images.

Figure 8 shows the visual results on the DIV2K dataset. The results clearly indicate that when the correct keys are used, the SSHR model successfully reveals high-quality secret images. The residual map, which is nearly entirely black, suggests minimal divergence from the ground truth. However, when an attacker uses incorrect keys, the system's security is evident, as no meaningful information can be extracted from the stego images. Even if the attacker intercepts the public key during transmission and attempts to decrypt the secret image, the model's security remains intact. Moreover, the model preserves the naturalness and imperceptibility of the stego images. These results collectively show that the suggested approach provides high security and efficacy.

Figure 9 illustrates the performance of our SSHR model compared to other generative steganography models on the UniStega dataset, using three different prompts. When compared to CRoSS (Yu et al., 2024) and DiffStega (Yang et al., 2024), our SSHR model significantly improves the naturalness and imperceptibility of the stego images. It also facilitates substantial modification of the secret image content, guided by the reference image, to minimize the similarity between the stego and secret image pairs. When incorrect keys are used during the reveal process, the security of our model is evident, as the exposed image differs drastically from the secret image and contains minimal secret information. Moreover, an attacker cannot recover any useful data if they attempt to expose the secret image using the intercepted public key. Additionally, our model almost perfectly recovers the secret image, as demonstrated by the near-black residual map between the recovered image and the ground-truth secret image, in stark contrast to the larger residuals found in CRoSS (Yu et al., 2024) and DiffStega (Yang et al., 2024). Visual results confirm that our SSHR model surpasses previous state-of-the-art (SOTA) models in terms of both effectiveness and security.

**Effectiveness of Wavelet Transform.** Following the success of prior work (Jing et al., 2021), we adopt Wavelet Transform to perform steganography in the frequency domain, converting the image from the spatial domain to the frequency domain. As shown in Table 5, Wavelet Transform significantly enhances our model, yielding a 2.59 dB improvement for reference/stego image pairs and a 0.88 dB improvement for revealed/secret image pairs. These results suggest that Wavelet Transform boosts performance at both the concealment and reveal stages, enhancing the quality of stego images while preserving the integrity of revealed secret images.

**Effectiveness of Rep-Conv.** As shown in Table 5, the Rep-Conv module significantly improves the quantitative metrics for both the stego and revealed images, with a 1.96 dB improvement for reference/stego image pairs and a 2.95 dB improvement for revealed/secret image pairs. These enhancements positively affect both the concealment and reveal stages, demonstrating that the Rep-Conv module effectively boosts the performance of the proposed model.

**Effectiveness of CIGM.** The CIGM preprocesses the reference images to optimize the generation process and gradually encrypts the secret images over time. As indicated in Table 5, the CIGM leads to substantial improvements in the quantitative metrics, providing a 1.4 dB improvement for reference/stego image pairs and a 3.37 dB improvement for revealed/secret image pairs. This enhancement enhances both the concealment and reveal stages. The results highlight that the CIGM reduces the similarity between the stego and secret images, while the high-quality recovered secret images further confirm the module's effectiveness and the overall performance of the model.

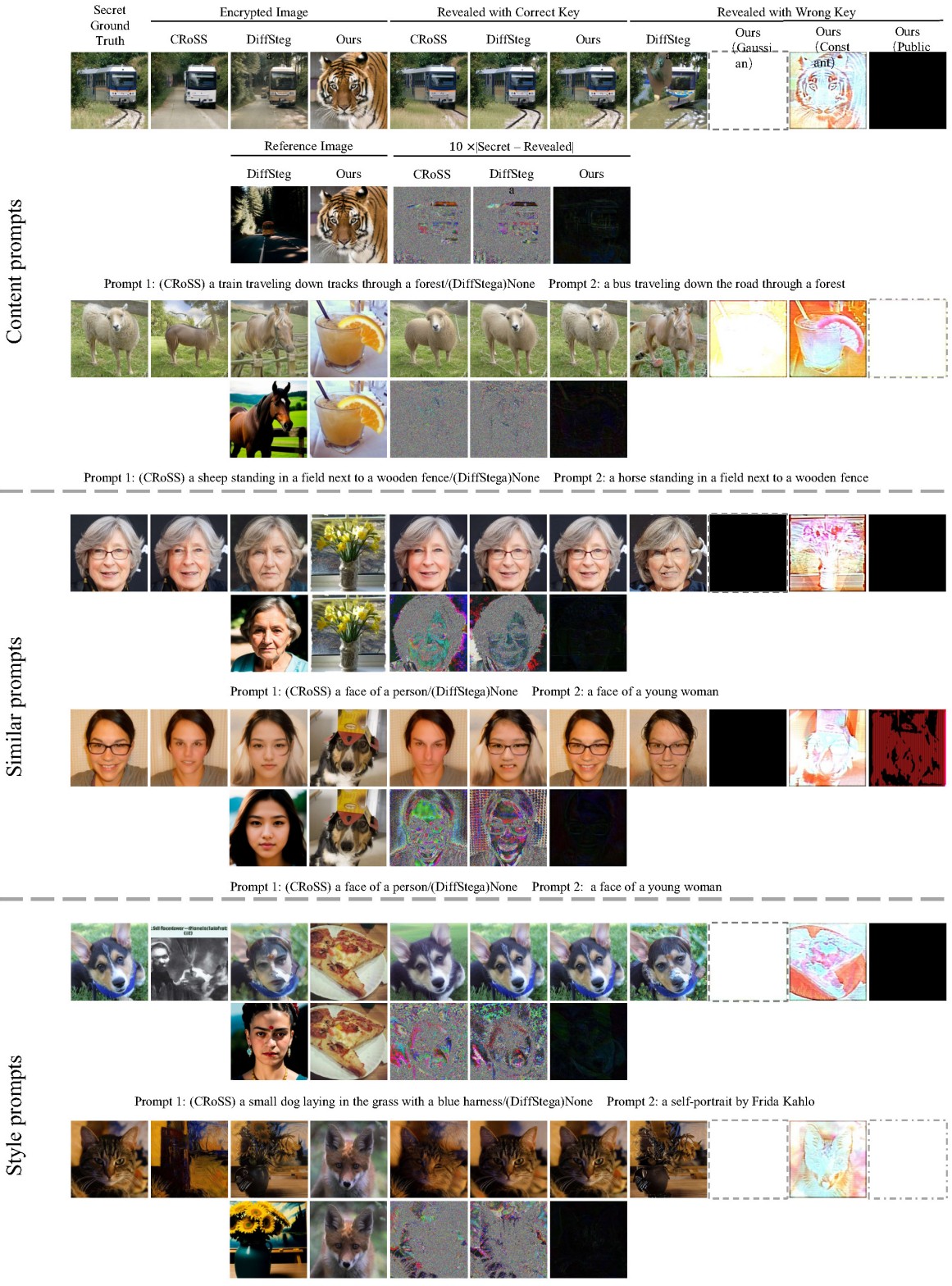

*Figure 9.* Visual contrast of our model and other generative steganography models on the UniStega dataset across three different prompts. These prompts are utilized in CRoSS and DiffStega, whereas our model functions without text prompts. The reference image is used as the image condition in both DiffStega and our model. The secret images are revealed with four types of keys: correct keys, constant, random noise and the public keys tied to secret images.

