# OpenReview forum: "SSHR: More Secure Generative Steganography with High-Quality Revealed Secret Images"
_ICML.cc/2025/Conference — ICML 2025 poster_

### Official Review · Reviewer_JH66 · 2025-03-11

**Overall Recommendation:** 3

**Summary:**

This paper proposes an image steganography method based on the diffusion model. By introducing a reference image and adaptive keys, it solves the problems of "limited control of text prompts" and "insufficient key security" in current methods, improves the quality of the revealed secret images, and enhances the system security. Experimental results show that the method proposed in this paper outperforms existing methods in terms of image quality and security.

## update after rebuttal
Thanks for authors' response; I will keep my score.

**Claims And Evidence:**

Yes, the claims in the paper are supported by clear experimental results.

**Essential References Not Discussed:**

None. All the major works in this field have been cited and discussed in this paper.

**Experimental Designs Or Analyses:**

The experimental design and analysis in this paper are reasonable.

**Methods And Evaluation Criteria:**

It is meaningful. The method proposed in this paper can significantly improve the image quality of steganography using diffusion models without the need for complex training. At the same time, this paper provides a new idea for the application of cryptography in steganography.

**Other Comments Or Suggestions:**

(1) A large number of formulas are unfriendly to readers. The authors should provide clear explanations for each process.
(2) What information needs to be shared between the sender and the receiver in advance? By reading the supplementary materials, it can be inferred that both parties share the same reference image, and the public key related to the secret image also needs to be sent together with the stego image. Such important information should be presented in the main paper rather than in the supplementary materials.

**Other Strengths And Weaknesses:**

Strengths
(1) Novel method design: This paper innovatively uses adaptive keys to control the conceal and exact reveal processes. Compared with previous methods relying on text prompts, it can improve the security of stego images.
(2) Excellent experimental results: Compared with other steganography methods based on diffusion models, the method proposed in this paper significantly improves the image quality.

Weaknesses
(1) Insufficiently clear description of the conceal process: In the description of the "Conceal process" section, this paper uses a large number of formulas but lacks deep explanations, which may cause reading difficulties for readers.
(2) Unclear experimental settings for steganalysis: In the "Steganographic analysis" experiment, there is a lack of explanations about the sample size of the experimental data and the steganographic covers.

**Questions For Authors:**

(1) Regarding Weakness 1, what processing steps does the secret image undergo after the DWT transformation? How exactly is the symmetric key used for encryption during the processing?
(2) In Paper [2], reference images were also used. However, in the experimental results, the stego images in [2] are significantly different from the reference images, while the stego images in this paper are almost indistinguishable from the reference images. Which designs in SSHR contribute to achieving the above results? Please provide a detailed analysis.
(3) Regarding Weakness 2, in the steganalysis experiment, the sample sizes of the training data and the test data are crucial and need to be stated in the experimental setup. Meanwhile, training a steganalysis detector requires "cover - stego" pairs. In generative steganography, it is necessary to clarify what the cover is (for example, in CRoSS [1], the stego image is directly generated from the secret image without a cover. So, what is the cover used in steganalysis?).
(4) In - depth discussion on reference images. In this paper, does the selection of reference images have a significant impact on the results? For example, are there obvious differences between the experimental results when the reference image is a real - world image and when it is a generated image?

**Relation To Broader Scientific Literature:**

This paper focuses on the steganography scenario of hiding images within images using diffusion models. Compared with previous related papers, it achieves higher image quality and system security.
In the papers of CRoSS [1] and DiffStega [2], text prompts are used as private keys. However, due to the limited control of text prompts, the quality of stego images is poor. This paper uses reference images and adaptive keys to avoid above problem, resulting in stego images with higher quality.
[1] Yu, Jiwen, et al. "Cross: Diffusion model makes controllable, robust and secure image steganography." Advances in Neural Information Processing Systems 36 (2023): 80730-80743.
[2] Yang, Yiwei, et al. "DiffStega: towards universal training-free coverless image steganography with diffusion models." arXiv preprint arXiv:2407.10459 (2024).

**Theoretical Claims:**

This paper does not involve theoretical proofs.

---

> ### Author Rebuttal · Authors · 2025-03-31
>
> We greatly appreciate your thorough feedback and the time you've dedicated to reviewing our work. We sincerely hope that our clarifications will address your concerns and strengthen your confidence in our work.
>
> A1: (1)  Conceal and Reveal Process. In the proposed model, the secret image $x_{sec}$ is initially taken as input and transformed into the latent space as $z_{sec}$  using the discrete wavelet transform. The latent representation  $z_{sec}$ is then encrypted with the symmetric key $k_{sym} $ and the reference image $x_{ref}$, which has been pre-processed through the condition information guidance module. The conceal process follows Equation (12), with the input $z_0 = z_{sec}$ and the output $z_T = z_{stego}$, generating the latent representation $z_{stego}$ of the stego image $x_{stego}$. The final stego image $x_{stego}$ is then obtained by applying the inverse wavelet transform. The reveal process mirrors the conceal process, following Equation (14) with input $z_T = z_{stego}$ and output $z_0 = z_{rev}$, ultimately yielding the revealed secret image $x_{rev}$.
>
> (2) Symmetric Key Usage. The symmetric key is employed to generate the weights for the Conditional Re-parameterization Convolution and the modulation parameters  $\alpha$, $\beta$, $\gamma$ in the Condition Reaction Term. This design ensures that the symmetric key plays a crucial role in the conceal and reveal process.
>
> Your constructive comments are both sincerely valued and deeply appreciated, and we will re-organize and add more clearer description in the future version.
>
> A2: The outcome of this model is influenced by: the removal of the text prompt and the design of the model’s training objectives.
>
> (1) The removal of the text prompt in the proposed model contributes to reducing the risk of semantic misalignment and improves control over specific semantic regions. The proposed model eliminates the text prompts entirely, relying solely on the reference images to guide the generation of stego images. This architectural simplification facilitates more natural stego images generation, reduces the risk of semantic misalignment and removes the need for balancing multi-modal prompts. In contrast, the DiffStega model integrates both reference images and text prompts to steer the generation of stego images. With the dual-modal guidance, DiffStega generates stego images that align with the constraints of text prompts and the visual features of the reference images. However, text prompts in generative steganography models offer inadequate control, leading to the trade-off between the two types of prompts.
>
> (2) The design of the training objective plays a crucial role in achieving the results outlined in this question, aligning with the discussed design choices. The framework’s focus on reference-image guidance and its simplification of the generative process contribute significantly to the system’s overall efficacy.
>
> A3: (1) Dataset Resolution. Both the training and test datasets are utilized at a resolution of 256$\times$256, as documented in the manuscript. To improve clarity and readability, the manuscript will be reorganized to provide a more structured presentation of the information.
>
> (2) Steganalysis Protocol. The deep learning steganalysis architecture follows the training protocol established by SRNet (Deep Residual Network for Steganalysis of Digital Images). After training the deep steganalysis network, we assess the anti-steganalysis capabilities of various methods with the trained network. This enables a comprehensive assessment of the model's anti-steganalysis capabilities. Our experimental setup aligns with common practices used in existing steganography research, guaranteeing the objectivity of the results. We aim to explore more effective experimental settings in future work to further enhance the evaluation of different steganography models.
>
> A4: This question closely resembles the first one of the second reviewer. With respect to the concerns you've raised, we will now provide a renewed explanation of this issue.
>
> (1) Reference Image Selection. We would like to clarify that the selection of reference images does not significantly affect the model's performance. During both training and testing, reference images are randomly selected, which enhances the model's generalization ability, ensuring excellent performance across various types of reference images. This makes the proposed model more adaptable to different visual contexts.
>
> (2) Model Performance. The proposed model consistently delivers high-quality stego-images and secure secret recovery, regardless of whether the reference images are real-world images or generated images.
>
> Your insightful question presents a promising avenue, selection of reference images in generative steganography,  for future exploration. We intend to investigate this aspect in future work to enhance the effectiveness and adaptability of generative steganography techniques.

---

### Official Review · Reviewer_c4WS · 2025-03-12

**Overall Recommendation:** 4

**Summary:**

The paper presents a novel generative steganography method, SSHR, which incorporates the diffusion model to address challenges in image steganography. It replaces the traditionally used text prompts with reference images and adaptive symmetric keys to generate stego images, providing greater control over the image generation process and enhancing the security and naturalness of the generated images. SSHR uses an Exact Reveal Process to improve the quality of revealed secret images and introduces a Reference-Secret Image Related Symmetric Key (RSRK) generation module to enhance the security of both the keys and the concealed secret images.

**Claims And Evidence:**

Clear and convincing.

**Essential References Not Discussed:**

The references are efficient.

**Experimental Designs Or Analyses:**

Yes, the experiments are reasonable.

**Methods And Evaluation Criteria:**

Yes.

**Other Comments Or Suggestions:**

N/A

**Other Strengths And Weaknesses:**

The approach introduced by the authors is highly original. The shift from text prompts to reference images for guiding stego image generation is a unique and creative solution. Additionally, the Exact Reveal Process and adaptive key generation offer new ways to improve the recovery of secret images and prevent unauthorized access.

**Questions For Authors:**

1. How does the proposed SSHR model perform when the reference images used for generation contain irrelevant or conflicting features with the secret images? Does the model still maintain high-quality stego images and secure secret recovery?
2. Has the symmetric key generation process been tested against potential vulnerabilities, such as key leakage in real-world scenarios? Are there any theoretical risks to this approach in terms of cryptanalysis?
3. While the exact reveal process is claimed to improve the quality of revealed images, has this been verified for large-scale datasets with real-world complexities (e.g., varying lighting conditions, image compression, etc.)?

**Relation To Broader Scientific Literature:**

The paper addresses a significant problem in generative steganography, where the quality and security of secret images are at risk. The innovative integration of reference images and adaptive keys improves both the imperceptibility and security of the generated stego images, making this method highly relevant for modern image privacy and security applications.

**Theoretical Claims:**

Correct.

---

> ### Author Rebuttal · Authors · 2025-03-31
>
> We greatly appreciate the very detailed feedback and your recognition of our contributions! We sincerely hope our response below will further enhance your confidence in our work.
>
>  A1: (1) Reference Image Selection and Model Performance. The selection of the reference image does not significantly impact the model’s performance. Reference images used for generative guidance in the proposed framework are randomly selected during both the training and testing phases. This stochastic selection mechanism ensures that the model demonstrates robust generalization across diverse reference images, maintaining effective steganographic performance even when the textures of the reference images are irrelevant or conflict with those of the secret images.
>
> (2) Model Performance. The proposed model continues to produce high-quality stego-images and ensures secure secret recovery. The stochastic selection mechanism guarantees robust generalization across various reference images, preserving high-fidelity stego-image synthesis and secure secret recovery, even when the reference images contain irrelevant or conflicting visual features in relation to the secret data.
>
> Furthermore, your insightful question holds significant value and highlights an important avenue for advancement. We plan to systematically investigate this direction to identify optimal reference image selection strategies that strike a balance between creative flexibility and security guarantees for generative image steganography frameworks in future work.
>
> A2: (1) Symmetric Key Security. The symmetric key’s security is rigorously maintained during data transmission. In the proposed framework, only the public key required for symmetric key derivation is exchanged, eliminating the risk of exposing the private key. This design ensures that the symmetric key's confidentiality is upheld by securely storing the private key, thereby significantly enhancing the overall security of the system.
>
> (2) Theoretical Security. The proposed cryptographic key generation process, grounded in robust, well-established algorithms, mitigates potential theoretical vulnerabilities. Specifically, the symmetric key generation method employs the Elliptic Curve Diffie-Hellman Ephemeral (ECDHE) algorithm. This choice ensures strong theoretical security guarantees, safeguarding the symmetric key’s generation and protection.
>
> (3) Experimental Results. a) Evaluation on Various Datasets. We evaluated the proposed model on four distinct real-world datasets: the DIV2K test dataset (100 images), COCO (5,000 images), ImageNet (10,000 images), and UniStega (100 images). These datasets, sourced from real-world scenarios, were chosen to assess the model’s performance and generalization across a variety of image types. The results highlight the model’s outstanding performance, demonstrating its robustness and adaptability in diverse real-world scenario applications.
>
> b) Evaluation on Various Keys. The proposed model was also tested with several types of keys, including constant keys, random Gaussian noise, the public key transmitted to the receiver, and the correct symmetric key. Experimental findings confirmed that third parties could not extract secret images from stego images when using incorrect keys. This result underscores the high level of security provided by the proposed key generation process in real-world scenarios.
>
> A3: Datasets. The proposed model was evaluated on a diverse range of real-world image datasets that cover various scenarios and challenging conditions, such as varying lighting, resolution differences, and environmental factors. To ensure a thorough assessment and facilitate a comprehensive comparison with other steganography approaches, including both cover-based and generative methods, we adhered to prior steganography methods. The model was tested across four distinct real-world datasets: the DIV2K test dataset (100 images), COCO (5,000 images), ImageNet (10,000 images), and UniStega (100 images). These datasets encompass a variety of image resolutions, scenarios, and visual contexts, providing a robust foundation for evaluating the model's performance under different conditions.

---

### Official Review · Reviewer_iwX7 · 2025-03-13

**Overall Recommendation:** 2

**Summary:**

This paper proposed a targeted solution to some drawbacks in diffusion model-based generative steganography with text prompts. Although various experiments indicate the proposed model can outperform existing methods in terms of recovery quality and secret image security, there still exist some issues：

1)	In theory: Although the authors state that SSHR is a generative steganography model, it introduces an additional reference image, and the goal of the model is to make the generated stego image similar to that reference one. Thus, its essence is hiding image in image, and it is not strictly coverless generative steganography.

2)	In technology: The proposed SSHR builds on previous work (Jing et al., 2021) and differs in that it is conducted in the frequency domain only. As for the innovations in generative modeling, it only introduce a condition term R(z, ksym, c) into the original PM diffusion model. In addition, the symmetric key is generated at the sender's end, so how does it be securely transmitted to the receiver?

3)	In experiments: Although the authors have conducted a comprehensive experimental validation of the quality assessment of the generated images, the experimental aspects regarding steganographic security are problematic. Since the method is essentially hiding image in image, the reference image should be used as a COVER and the containing image should be used as a STEGO. In addition to using the deep learning-based steganalysis tools in the paper, a handcrafted feature-based steganalyzer should be considered for
detection.

4)	In writing: There are some typos: “Peivate”, “within within”,…

**Claims And Evidence:**

Yes

**Essential References Not Discussed:**

No

**Experimental Designs Or Analyses:**

Although the authors have conducted a comprehensive experimental validation of the quality assessment of the generated images, the experimental aspects regarding steganographic security are problematic. Since the method is essentially hiding image in image, the reference image should be used as a COVER and the containing image be used as a STEGO. In addition to using the deep learning-based steganalysis tools in the paper, a handcrafted feature-based steganalyzer should be considered for detection.

**Methods And Evaluation Criteria:**

Partly

**Other Comments Or Suggestions:**

No

**Other Strengths And Weaknesses:**

See Summary

**Questions For Authors:**

See Summary

**Relation To Broader Scientific Literature:**

N/A

**Theoretical Claims:**

No

---

> ### Author Rebuttal · Authors · 2025-04-01
>
> We greatly appreciate your valuable feedback and sincerely hope our response adequately addresses your points and restores your confidence in our work.
>
> A1: Fundamental Disparities from Cover-based Steganography.  We clarify that the proposed method effectively bridges the gap between cover-based steganography and coverless generative steganography, markedly diverging from cover-based methods that directly embed secret data into cover images. The proposed method adheres to a generative steganography pipeline, with reference images serving as guidance for generating stego images. It offers a hybrid solution that unites cover-based steganography and coverless generative steganography. Unlike coverless methods, the proposed method exploits reference images to boost stego images' naturalness and imperceptibility.
>
> A2: (1) Fundamental Disparities from HiNet. The proposed model presents a significant departure from HiNet (Jing et al., 2021), a cover-based method. The key distinctions are: a) Generative steganography vs Cover-Based steganography. The proposed method follows a generative steganography pipeline, with reference images serving as guidance for the generation of stego images. This distinguishes the proposed method from HiNet, which directly embeds secret data into cover images; b) Key Management. The proposed model dynamically derives secret-specific symmetric keys through public key exchange and shared symmetric key generation module. Conversely, HiNet lacks mechanisms for key management; c) Role of Auxiliary Images. Although both the proposed method and HiNet utilize auxiliary images, the functional roles differ fundamentally. In the proposed method, the reference images serve as image prompts and guide the stego images generation process, whereas HiNet treats secondary images as secret data containers.
>
> (2) Main Contributions. The proposed method addresses several fundamental challenges in image steganography, including naturalness and imperceptibility, quality of the revealed secret images and security. Unlike modular approaches that rely on isolated components, our method integrates three main contributions: a) We systematically propose a novel generative steganography method joints the reference images with the adaptive keys to govern the entire steganography process, enhancing the naturalness and imperceptibility of the stego images; b) We methodically design an Exact Reveal Process to precisely reverse the conceal process, minimizing errors in the reveal phase and improving the quality of the revealed secret images; c) We propose a Reference-Secret Image Related Symmetric Key generation module for dynamic symmetric keys generation, bolstering the security of both the keys and the secret images. It is important to note that these challenges cannot be adequately resolved with any singular modular component operating in isolation. The proposed model achieves optimal performance only through the synergy of these contributions.
>
> (3) Key Transmission. The symmetric key is not transmitted directly to the receiver. Only the public keys associated with the secret images are delivered. a) Public Key Exchange. The sender generates a public-private key pair and transmits the public key to the receiver alongside the stego image; b) Symmetric Key Derivation. The receiver uses the received public key within the shared symmetric key generation module to derive the symmetric key. This process decouples the symmetric key transmission from public key distribution and ensures key consistency without transmitting the symmetric key. Please refer to Section $\textbf{\textit{Reference-Secret Image Related Key}}$ and the supplementary materials $\textbf{\textit{Security of the Symmetric Key}}$ for detailed specifications of the transmission of keys and symmetric key derivation.
>
> A3:  (1) Experimental Setting. The proposed model presents a novel generative steganography method, distinguishing itself from cover-based methods. It bridges cover-based steganography and coverless generative steganography, and utilizes reference images as guidance for stego images generation rather than containers as in cover-based models. This fundamental divergence shapes the experimental design, which integrates elements of both cover-based and generative steganography, providing a unique and comprehensive evaluation framework.
>
>  (2) Steganalysis. We have employed the handcrafted feature-based steganalyzer to assess the anti-steganalysis capabilities of the proposed model. We utilize StegExpose, an open-source steganalysis tool integrates four handcrafted feature-based steganalyzers (Sample Pairs, RS Analysis, Chi-Square Attack and Primary Sets), for assessment. The experimental results, detailed in Section $\textbf{\textit{Steganographic Analysis}}$, demonstrate that the proposed model exhibits excellent resistance to steganalysis.
>
> A4: Thanks for mentioning the typos in our manuscript. We will review and polish the entire manuscript.

---

> > ### Comment · Reviewer_iwX7 · 2025-04-02
> >
> > （1）As you claimed, the  public key should be transmitted to the receiver alongside the stego image,  so how can the security of the public key transmission be ensured?
> >
> > （2） StegExpose is outdated for a long time, you should use other SOTA ones, e.g., SRM + Ensemble.

---

> > > ### Author Response · Authors · 2025-04-03
> > >
> > > Thank you for taking the time to provide additional feedback. We sincerely hope the following clarifications could address your points.
> > >
> > > A1: Public keys are designed for public transmission, representing a core tenet of key exchange protocols.
> > >
> > > The proposed symmetric key generation module is derived from the Elliptic Curve Diffie-Hellman Ephemeral (ECDHE) [1] protocol. Within cryptographic key exchange protocols, security of the symmetric key is predicated not upon public key confidentiality, but rather on two main principles:
> > >
> > > a) Mathematical Security Foundations. The security of key exchange protocols (e.g. RSA, ECDH) is mathematically grounded in computationally hard problems like integer factorization for RSA, and elliptic curve discrete logarithm problem for ECDH and so on. These foundational problems guarantee the computational infeasibility of deriving private keys or shared secrets from publicly available parameters.
> > >
> > > b) Security Architecture. The security of symmetric keys and overall system integrity guarantees from: (1) the computational intractability of asymmetric mathematical problems; (2) ephemeral session key generation ensuring forward secrecy.
> > >
> > > Key exchange protocols leverage sophisticated mathematical foundations to obviating the requirement for confidential public key dissemination.
> > >
> > > This study pioneers the integration of cryptographic key exchange protocols into image steganographic systems, establishing enhanced theoretical and practical security assurances for both encryption keys and concealed image data. To the best of our knowledge, this is the first exploration to integrate key exchange protocols into image steganography.
> > >
> > > A2: Following the common practices in the cover-based and generative steganography task (e.g., ISN [2], HiNet [3], CRoSS [4], DiffStega [5]), our evaluation framework employs StegExpose, XuNet, and SRNet, to systematically evaluate the anti-steganalysis capabilities of various models. The evaluation framework doesn't contain the SRM method, and this is justified by:
> > >
> > > a) The evaluation framework can provide a comprehensive evaluation of anti-steganalysis capabilities. The proposed model undergoes rigorous comparative evaluation against SOTA steganography models, involving both cover-based and generative models. All evaluated steganography models undergo systematic assessment with both classical statistical steganalysis methods (StegExpose) and deep learning steganalysis models (XuNet/SRNet), to assess their anti-steganalysis capabilities. This multidimensional evaluation framework enables rigorous and comprehensive evaluation of anti-steganalysis capabilities across various models.
> > >
> > > b) The evaluation framework achieves higher steganalysis accuracy relative to conventional SRM. Although SRNet, XuNet, and SRM all utilize noise residual computation and classification architectures, empirical evidence demonstrates that XuNet [6] and SRNet [7] achieve superior steganalysis performance compared to conventional SRM. This enhancement stems from their superior multi-scale feature extraction capabilities and superior noise residual characterization. The empirical evidence justifies prioritizing contemporary deep neural architectures exhibiting superior compatibility with modern steganography frameworks. Consequently, the SRM has not been contained in the evaluation framework.
> > >
> > > Your insightful question highlights a critical methodological consideration. While current time constraints preclude immediate integration of the suggested steganalysis approach, we commit to its systematic implementation in both the final manuscript and subsequent research.
> > >
> > > [1] Mehibel N, Hamadouche M H. A new approach of elliptic curve Diffie-Hellman key exchange[C]//2017 5th International Conference on Electrical Engineering-Boumerdes. IEEE, 2017: 1-6.
> > >
> > > [2] Lu S P, Wang R, Zhong T, et al. Large-capacity image steganography based on invertible neural networks[C]//Proceedings of the IEEE/CVF conference on computer vision and pattern recognition. 2021: 10816-10825.
> > >
> > > [3] Jing J, Deng X, Xu M, et al. Hinet: Deep image hiding by invertible network[C]//Proceedings of the IEEE/CVF international conference on computer vision. 2021: 4733-4742.
> > >
> > > [4] Yu J, Zhang X, Xu Y, et al. Cross: Diffusion model makes controllable, robust and secure image steganography[J]. Advances in Neural Information Processing Systems, 2023, 36: 80730-80743.
> > >
> > > [5] Yang Y, Liu Z, Jia J, et al. DiffStega: towards universal training-free coverless image steganography with diffusion models[C]//Proceedings of the Thirty-Third International Joint Conference on Artificial Intelligence. 2024: 1579-1587.
> > >
> > > [6] Xu G, Wu H Z, Shi Y Q. Structural design of convolutional neural networks for steganalysis[J]. IEEE Signal Processing Letters, 2016, 23(5): 708-712.
> > >
> > > [7] Boroumand M, Chen M, Fridrich J. Deep residual network for steganalysis of digital images[J]. IEEE Transactions on Information Forensics and Security, 2018, 14(5): 1181-1193.

---

### Decision · Program_Chairs · 2025-05-01

**Decision:**

Accept (poster)

**Comment:**

This paper proposes a generative steganography method to address challenges in image steganography. The paper originally received 1xWeakReject, 1xWeakAccept, and 1xAccept. The main concerns include limited novelty, insufficient experiments, symmetric key security, unclear statements, etc. The authors have provided rebuttals. Afterward, all reviewers keep their ratings. In the discussion phase, Reviewer iwX7 still keeps negative about the security of the public key transmission and evaluation framework. The authors have clarified these concerns in their second rebuttal. Although Reviewer iwX7 does not respond, the authors’ rebuttal addresses the reviewer’s concerns. Considering the rebuttal and discussions from all reviewers, ACs recommend accepting this paper. The authors are suggested to carefully revise the paper and incorporate newly conducted experiments according to the comments and discussions.